# Provably Sample-Efficient Active Preference Data Collection

## Abstract

Collecting human preference feedback is often expensive, leading recent works to develop algorithms to select them more efficiently. However, these works assume that the underlying reward function is linear, an assumption that does not hold in many real-life applications, e.g., online recommendation. To address this limitation, we propose `Neural-ADB`, an algorithm based on the neural contextual dueling bandit framework that provides a practical method for collecting human preference feedback when the underlying latent reward function is non-linear. We theoretically show that when preference feedback follows the Bradley-Terry-Luce model, the worst sub-optimality gap of the policy learned by `Neural-ADB` decreases at a sub-linear rate as the preference dataset increases. Our experimental results on preference datasets further corroborate the effectiveness of `Neural-ADB`.

## 1 Introduction

Collecting human preference feedback is essential in many real-life applications, like online recommendations (Kohli et al., 2013; Wu et al., 2023; Zhang and Wang, 2023; Yang et al., 2024), content moderation (Avadhanula et al., 2022), medical treatment design (Lai and Robbins, 1985; Bengs et al., 2021), prompt optimization (Lin et al., 2024), and aligning large language models (Bai et al., 2022; Menick et al., 2022; Mehta et al., 2023; Chaudhari et al., 2024; Das et al., 2024; Ji et al., 2024), to ensure systems effectively align with user preferences and exhibit desired behaviors. However, this process is often costly due to the need for skilled evaluators, the complexity of tasks, and the time-intensive nature of producing high-quality, reliable human feedback. To address the challenge of balancing cost and effectiveness in aligning systems, this paper proposes principled and practical algorithms for efficiently collecting human feedback sequentially and adaptively to achieve the desired system behavior. Specifically, we aim to answer the following fundamental question:
***How to achieve desired system behavior while using as minimum human feedback as possible?***

Recent works (Mehta et al., 2023; Das et al., 2024) have modeled the problem of active human feedback collection as an active version of the contextual dueling bandit problem (ADB for brevity) (Saha, 2021; Bengs et al., 2022; Li et al., 2024), where context-arm pair in the contextual dueling bandits corresponds to a task for which human preference feedback is collected and then proposed algorithms to select context-arm pairs for human feedback sequentially and adaptively by exploiting collected preference dataset, i.e., past context-arm pairs with their preference feedback. The preference feedback between two context-arm pairs is commonly assumed to follow the Bradley-Terry-Luce (BTL) model[1] (Hunter, 2004; Bengs et al., 2022; Li et al., 2024; Lin et al., 2024; Verma et al., 2025) in which the probability of preferring a context-arm pair over others is proportional to the exponential of its reward. In many real-life applications, the number of context-arm pairs (e.g., user-movie pair in online movie recommendation) can be large or even infinite. Therefore, the reward for each context-arm pair is assumed to be an unknown function of its feature vector, such as a linear function (Mehta et al., 2023; Das et al., 2024).

To better align the system for optimal performance, we consider two key components: *context selection* and *arm selection*. The context selection aims to encourage diversity by exploring the context space, such as selecting prompts as diverse as possible in prompt optimization. Whereas

---

[1]For more than two context-arm pairs, preferences are typically modeled using the Plackett-Luce model (Soufiani et al., 2014).

arm selection focuses on identifying the arms that help learn the best arm for each context, such as selecting the most effective pair of responses to a given prompt that maximizes the system's learning (Lin et al., 2024; Verma et al., 2025). Since the goal is to identify the best arm for each context, selecting suboptimal arms provides less useful information than choosing better arms. Existing methods for active contextual dueling bandits (Mehta et al., 2023; Das et al., 2024) fail to incorporate an efficient arm selection strategy during the data collection process, thereby limiting the ability of these methods to achieve optimal performance.

An efficient arm selection strategy requires estimating the reward function to guide the arm selection process effectively. Since the reward function may not always be linear in practice, this paper parameterizes the reward function via a *non-linear function*, which needs to be estimated from the available preference dataset by using methods like Gaussian processes (Williams and Rasmussen, 2006; Srinivas et al., 2010) or neural networks (Zhou et al., 2020; Zhang et al., 2021). However, Gaussian processes have limited expressive power and fail to optimize highly complex functions. In contrast, neural networks (NNs) have greater expressive power, making them well-suited for modeling complex functions (Dai et al., 2023; Lin et al., 2023; 2024; Verma et al., 2025).

In this paper, we propose a neural active contextual dueling bandit algorithm, `Neural-ADB`, which uses an NN to estimate the unknown reward function using the available preference dataset. The context selection in `Neural-ADB` is adapted from Das et al. (2024), while arm selection strategies are based on, respectively, upper confidence bound (UCB) and Thompson sampling (TS), and adapted from Verma et al. (2025). Due to the differences in context selection strategy, arm selection strategies, and the use of a non-linear reward function, our theoretical analysis is completely different than related existing work (Mehta et al., 2023; Das et al., 2024). One of the key theoretical contributions of this paper is *providing an upper bound on the maximum Mahalanobis norm* of a vector from the fixed input space, measured with respect to the inverse of a positive definite Gram matrix that is constructed using finite, adapted samples from that space. Building on this result, we prove that the worst sub-optimality gap (defined in Eq. (1)) of the policy learned by `Neural-ADB` decreases at a sub-linear rate as the preference dataset size increases.

Specifically, our key contributions can be summarized as follows:

- We introduce the setting of active contextual dueling bandits with a non-linear reward function in Section 2. In Section 3, we propose a neural active contextual dueling bandit algorithm, `Neural-ADB`, which uses an NN to estimate the unknown reward function from the available preference dataset and then uses this estimate into the arm selection strategies.

- We prove an upper bound on the maximum Mahalanobis norm of a vector from the fixed input space, as measured with respect to the inverse of a positive definite Gram matrix (Theorem 1), where the gram matrix is constructed using finite, adapted samples from that input space. We show that this upper bound decays at a sub-linear rate as the number of samples used in the Gram matrix increases. This theoretical result itself is of independent interest, as it gives valuable insights beyond the specific application of our work.

- We prove that the worst sub-optimality gap of the policy learned by `Neural-ADB` with both of our arm selection strategies (Theorem 2 and Theorem 3) decreases at a sub-linear rate with respect to the size of preference dataset, specifically at rate of $\tilde{O}((\tilde{d}/T)^{\frac{1}{2}})$, where $\tilde{O}$ hides the logarithmic factors and constants, and $\tilde{d}$ is the effective dimension of context-arm feature vectors. The decay rate of the worst sub-optimality gap for `Neural-ADB` improves by a factor of $\tilde{O}((\tilde{d}\log T)^{\frac{1}{2}})$ compared to exiting algorithms (Mehta et al., 2023; Das et al., 2024), thus bridging the gap between theory and practice.

- Finally, in Section 4, our experimental results further validate the different performance aspects of `Neural-ADB`, highlighting its sample efficiency for preference data collection.

## 2 PROBLEM SETTING

We model active human preference feedback collection as an active contextual dueling bandit problem, where a labeler (human or simulator) provides preference feedback for a chosen pair of arms.

**Active contextual dueling bandit.** We consider an active contextual dueling bandit problem, where the underlying latent reward function can be non-linear. In each iteration of this problem, the learner's goal is to select a triplet containing a context and two arms for collecting preference feedback from a

labeler/human such that the collected preference dataset leads to superior performance. Let $\mathcal{C}$ be the set of contexts and $\mathcal{A}$ be the set of all possible arms. In each iteration, the learner selects a context $c_t \in \mathcal{C}$ and then two arms (denoted as $a_{t,1}$ and $a_{t,2}$) from the set of arms $\mathcal{A}$. After selecting the triplet of context and two arms, the learner receives a stochastic preference feedback $y_t$, where $y_t = 1$ implies the arm $a_{t,1}$ is preferred over arm $a_{t,2}$ for the context $c_t$ and $y_t = 0$ otherwise. We use $\varphi(c_t, a)$ to denote the context-arm feature vector for context $c_t$ and an arm $a$, where $\varphi : \mathcal{C} \times \mathcal{A} \to \mathbb{R}^d$ is a known feature map, such as one that concatenates the context and arm features.

**Preference model.** Following the dueling bandits literature (Saha, 2021; Bengs et al., 2022; Li et al., 2024; Verma et al., 2025), we assume the preference feedback follows the Bradley-Terry-Luce (BTL) model[2] (Hunter, 2004; Luce, 2005). Under the BTL preference model, the preference feedback has a Bernoulli distribution, where the probability that the first selected arm $a_{t,1}$ is preferred over the second selected arm $a_{t,2}$ for the given context $c_t$ is given by

$$\mathbb{P}\{a_{t,1} \succ a_{t,2}\} = \mathbb{P}\{y_t = 1 | c_t, a_{t,1}, a_{t,2}\} = \mu\left(f(\varphi(c_t, a_{t,1})) - f(\varphi(c_t, a_{t,2}))\right),$$

where $a_{t,1} \succ a_{t,2}$ used for brevity and denotes that $a_{t,1}$ is preferred over $a_{t,2}$ for the given context $c_t$, $\mu(x) = 1/(1 + e^{-x})$ is the sigmoid function, $f : \mathbb{R}^d \to \mathbb{R}$ is an unknown non-linear bounded reward function, and $f(\varphi(c, a))$ is the latent reward of the arm $a$ for the context $c$. We require the following standard assumptions on the function $\mu$ (commonly referred to as a *link function* in the bandit literature (Li et al., 2017; Bengs et al., 2022)):

**Assumption 1.**
- *Let* $\kappa_\mu \doteq \inf_{c \in \mathcal{C}, a, b \in \mathcal{A}} \dot{\mu}(f(\varphi(c, a)) - f(\varphi(c, b))) > 0$ *for all triplets of context* $(c)$ *and pair of arms* $(a, b)$.
- *The link function* $\mu : \mathbb{R} \to [0, 1]$ *is continuously differentiable and Lipschitz with constant* $L_\mu$*. For logistic function, we have* $L_\mu \le 1/4$.

**Performance measure.** We denote the collected preference dataset up to $T$ iterations by $\mathcal{D}_T = \{(c_s, a_{s,w}, a_{s,l}, y_s)\}_{s=1}^T$, where $a_{s,w} \succ a_{s,l}$ for the selected context $c_s$ in iteration $s$. We aim to learn a policy, $\pi : \mathcal{C} \to \mathcal{A}$ from the collected preference dataset $\mathcal{D}_T$ that achieves the worst sub-optimality gap across all contexts in $\mathcal{C}$, which is defined as follows:

$$\Delta_{\mathcal{D}_T}^\pi = \max_{c \in \mathcal{C}} \left[ \max_{a \in \mathcal{A}} f(\varphi(c, a)) - f(\varphi(c, \pi(c))) \right], \tag{1}$$

where policy $\pi$ is a learned policy from the collected preference dataset $\mathcal{D}_T$ up to the iteration $T$. The policy $\pi_{\mathcal{D}_T}$ competes with the Condorcet winner (Bengs et al., 2021; Das et al., 2024) for a given context, i.e., an arm that is better than all other arms. The suboptimality gap is the worst possible difference in latent rewards over the set of contexts, and the same performance measure is used in prior work (Mehta et al., 2023; Das et al., 2024).

## 3 Algorithm for Active Human Preference Feedback Collection

In this section, we introduce `Neural-ADB`, a simple yet principled and practical algorithm designed to efficiently select context-arm pairs for collecting preference feedback. `Neural-ADB` consists of two main components: context selection and arm selection. Since the arm selection strategy depends on the estimated reward function, we first explain how an NN can be used to estimate the unknown reward function. We will then give details of the context and arm selection strategies, followed by our theoretical results that validate the effectiveness of `Neural-ADB`.

### 3.1 Reward function estimation using neural network

For estimating the latent reward function, we use a fully connected neural network (NN) with depth $D \ge 2$, a hidden layer width $w$, and ReLU activations as done in Zhou et al. (2020), Zhang et al. (2021), and Verma et al. (2025). Let $h(x; \theta)$ be the output of a full-connected NN with parameters $\theta$ for context-arm feature vector $x = \varphi(c, a)$ of context $c$ and arm $a$, which we define as follows:

$$h(x; \theta) = \boldsymbol{W}_D \text{ReLU}\left(\boldsymbol{W}_{D-1} \text{ReLU}\left(\cdots \text{ReLU}\left(\boldsymbol{W}_1 x\right)\right)\right),$$

---

[2]Our results are also applicable to any preference models, such as the Thurstone-Mosteller model and Exponential Noise, as long as stochastic transitivity holds (Bengs et al., 2022).

where $\text{ReLU}(v) = \max\{v, 0\}$, $\boldsymbol{W}_1 \in \mathbb{R}^{w \times d}$, $\boldsymbol{W}_l \in \mathbb{R}^{w \times w}$ for $2 \le l < D$, $\boldsymbol{W}_D \in \mathbb{R}^{w \times 1}$. The parameters of the NN are represented by $\theta = (\text{vec}(\boldsymbol{W}_1); \cdots \text{vec}(\boldsymbol{W}_D))$, where $\text{vec}(A)$ transforms an $m \times n$ matrix $A$ into a vector of dimension $mn$. We use $p$ to represent the total number of NN parameters, which is given by $p = dw + w^2(D-1) + w$, and $g(x; \theta)$ to denote the gradient of NN $h(x; \theta)$ with respect to $\theta$. At the end of each iteration $t$, the preference dataset $\mathcal{D}_t = \{(c_s, a_{s,w}, a_{s,l}, y_s)\}_{s=1}^t$ is used to estimate the reward function $f$ by training an NN $h$ (parameterized by $\theta_{t+1}$) using gradient descent to minimize the following binary cross entropy loss function:

$$\min_\theta \mathcal{L}_t(\theta) = -\frac{1}{w} \sum_{s=1}^t \Big[ \log \mu\big(h(\varphi(c_s, a_{s,w}); \theta) - h(\varphi(c_s, a_{s,l}); \theta))\big) \Big] + \frac{1}{2}\lambda \|\theta - \theta_0\|_2^2, \quad (2)$$

where $\theta_0$ denotes the initial parameter of the NN that is initialized according to the standard practice in neural bandits (Zhou et al., 2020; Zhang et al., 2021) (see Algorithm 1 in Zhang et al. (2021) for details). Minimizing the first term in the above loss function (that involves the summation over the $t$ terms) corresponds to finding the maximum log-likelihood estimate of the parameters $\theta$.

## 3.2 Neural-ADB

We next propose a simple yet principled and practical algorithm, Neural-ADB, that consists of two key components: Context selection and arm selection. Neural-ADB works as follows: At the beginning of the iteration $t$, we first select the context as follows:

$$c_t = \underset{c \in \mathcal{C}}{\text{argmax}} \max_{(a,b) \in \mathcal{A} \times \mathcal{A}} \|\Phi(c, a) - \Phi(c, b)\|_{V_t^{-1}}, \quad (3)$$

where $\Phi : \mathcal{C} \times \mathcal{A} \to \mathbb{R}^p$ is a known feature map and $V_{t-1} = \frac{\lambda}{\kappa_\mu}\mathbb{I}_p + \sum_{s=1}^{t-1} z_s z_s^\top \frac{1}{w}$ in which $z_s = \Phi(c_s, a_{s,w}) - \Phi(c_s, a_{s,l}) = g(\varphi(c_s, a_{s,w}); \theta_0) - g(\varphi(c_s, a_{s,l}); \theta_0)$, and $g(\varphi(c_s, a_{s,i}); \theta_0)/\sqrt{w}$ is used as the Random features approximation for the context-arm feature vector $\varphi(c_s, a_{s,i})$. This strategy is adapted from the context selection strategy[3] from Das et al. (2024). After selecting context $c_t$, Neural-ADB uses the trained NN (as an estimate of the unknown reward function) to decide which two arms must be selected. To do so, Neural-ADB uses UCB- and TS-based arm selection strategies, which efficiently balance the trade-off between exploration and exploitation (Lattimore and Szepesvári, 2020) due to the bandit nature of preference feedback, as preference feedback is only observed for the selected pair of arms.

---

**Neural-ADB** Neural Active Dueling Bandit algorithm

---

1: **Input parameters:** $\delta \in (0, 1)$, $\lambda > 0$, and $w > 0$
2: **Initialize:** NN parameters $\theta_1$ and $D_0 = \emptyset$
3: **for** $t = 1, \ldots, T$ **do**
4:     Select a context $c_t$ from $\mathcal{C}$ using Eq. (3)
5:     Select first arm $a_{t,1}$ using Eq. (4)
6:     Select second arm $a_{t,2}$ using Eq. (5) (UCB-based selection) or Eq. (7) (TS-based selection)
7:     Observe preference feedback $y_t = \mathbb{1}_{\{a_{t,1} \succ a_{t,2}\}}$
8:     Update $D_t = D_{t-1} \cup \{(c_t, a_{t,1}, a_{t,2}, y_t)\}$
9:     Retrain NN parameters $\theta_{t+1}$ using $D_t$ by minimizing the loss function defined in Eq. (2)
10: **end for**
11: Return policy $\pi(c) = \underset{a \in \mathcal{A}}{\text{argmax}}\, h(\varphi(c, a); \theta_T), \forall c \in \mathcal{C}$

---

**UCB-based arm selection strategy.** Algorithms based on Upper confidence bound (UCB) are commonly used to address the exploration-exploitation trade-off in many sequential decision-making problems (Auer et al., 2002; Abbasi-Yadkori et al., 2011; Zhou et al., 2020; Bengs et al., 2022). Our UCB-based arm selection strategy works as follows: In the iteration $t$, it selects the first arm greedily (i.e., by maximizing the output of the trained NN with parameters $\theta_t$) for the selected context $c_t$, ensuring the best-performing arm is always selected as follows:

$$a_{t,1} = \underset{a \in \mathcal{A}}{\text{argmax}}\, h(\varphi(c_t, a); \theta_t). \quad (4)$$

---

[3]Selecting contexts uniformly at random suffer a constant sub-optimality gap (Das et al., 2024, Theorem 3.2).

The second arm $a_{t,2}$ is selected optimistically by maximizing the UCB value as follows:

$$a_{t,2} = \underset{b \in \mathcal{A} \setminus \{a_{t,1}\}}{\mathrm{argmax}} \left[ h(\varphi(c_t, b); \theta_t) + \mathrm{cf}(t, c_t, a_{t,1}, b) \right], \tag{5}$$

where $\mathrm{cf}(t, c_t, a_{t,1}, b) = \nu_T \sigma_{t-1}(c_t, a_{t,1}, b)$, $\nu_T \doteq (\beta_T + B\sqrt{\lambda/\kappa_\mu} + 1)\sqrt{\kappa_\mu/\lambda}$ in which $\beta_T \doteq \frac{1}{\kappa_\mu}\sqrt{\widetilde{d} + 2\log(1/\delta)}$, $\widetilde{d}$ is the *effective dimension* (defined in Eq. (8)), and

$$\sigma_{t-1}^2(c, a, b) \doteq \frac{\lambda}{\kappa_\mu} \left\| \frac{1}{\sqrt{w}}(\varphi(c, a) - \varphi(c, b)) \right\|_{V_{t-1}^{-1}}^2. \tag{6}$$

A larger value of $\sigma_{t-1}^2(c_t, a_{t,1}, b)$ implies that arm $b$ is significantly different from $a_{t,1}$, given the contexts and arm pairs already selected. As a result, the second term in Eq. (5) makes the second arm different from the first arm which ensures exploration.

**TS-based arm selection strategy.** Thompson sampling (TS) selects an arm based on its probability of being the best (Thompson, 1933). Several works (Chapelle and Li, 2011; Agrawal and Goyal, 2013; Chowdhury and Gopalan, 2017; Li et al., 2024) have shown that TS empirically outperforms UCB-based bandit algorithms. Therefore, we also propose a TS-based arm selection strategy in which the first arm is also selected using Eq. (4) and the second arm is selected differently. To select the second arm, it first samples a score $s_t(b) \sim \mathcal{N}\big(h(\varphi(c_t, b); \theta_t) - h(\varphi(c_t, a_{t,1}); \theta_t), \nu_T^2 \sigma_{t-1}^2(c_t, a_{t,1}, b)\big)$ for every arm $b \in \mathcal{A} \setminus \{a_{t,1}\}$ and then selects the second arm that maximizes the samples scores as follows:

$$a_{t,2} = \mathrm{argmax}_{b \in \mathcal{A} \setminus \{a_{t,1}\}} s_t(b). \tag{7}$$

After selecting context and arms in iteration $t$, stochastic preference feedback is observed, denoted by $y_s = \mathbb{1}_{\{a_{t,1} \succ a_{t,2}\}}$, which is equal to 1 if arm $a_{s,1}$ is preferred over arm $a_{s,2}$ for context $c_t$ and 0 otherwise. With the new observation, the preference dataset is updated to $\mathcal{D}_t = \mathcal{D}_{t-1} \cup \{(c_t, a_{t,w}, a_{t,l}, y_t)\}$ and then the NN is retrained using the updated preference dataset $\mathcal{D}_t$. Once the preference data collection process concludes (i.e., end of iteration $T$, which may not be fixed a priori), Neural-ADB returns the following policy: $\forall c \in \mathcal{C} : \pi(c) = \mathrm{argmax}_{a \in \mathcal{A}} h(\varphi(c, a); \theta_T)$.

## 3.3 THEORETICAL RESULTS

Let the number of arms in $\mathcal{A}$ be finite, and define $\mathbb{V} = \sum_{s=1}^{T} \sum_{(a,b) \in \mathcal{A} \times \mathcal{A}} z_{a,b}(s) z_{a,b}(s)^\top \frac{1}{w}$, where $z_{a,b}(s) = \varphi(c_s, a) - \varphi(c_s, b)$ and $C_2^{|\mathcal{A}|}$ denotes all pairwise combinations of arms. Then, the *effective dimension* of context-arm feature vectors is defined as follows:

$$\widetilde{d} = \log \det \left( \frac{\kappa_\mu}{\lambda} \mathbb{V} + \mathbb{I}_p \right). \tag{8}$$

In the following, we present a novel theoretical result that gives an upper bound on the maximum Mahalanobis norm of a vector selected from the fixed input space, measured with respect to the inverse of a positive definite Gram matrix constructed from finite, adapted samples of the same space.

**Theorem 1.** *Let $\{Z_s = z_s z_s^\top\}_{s=1}^{T}$ be a finite adapted sequence of self-adjoint matrices in $\mathbb{R}^d$. Define $\mathbb{E}\left[z_s z_s^\top\right] = \Sigma_s \leq \Sigma_{\max}$, $V_0 = \lambda \mathbb{I}_d$, $V_T = \lambda \mathbb{I}_d + \sum_{s=1}^{T} z_s z_s^\top$. Assume $\|z_s\|_2 \leq L$ for all $z \in \mathcal{Z} \subset \mathbb{R}^d$, $\lambda_{\min}(A)$ denote the minimum eigenvalue of a matrix $A$, and $\forall s \leq T :$ $\|V_s - V_{s-1}\|^2 \leq C_s$, where $\|V\|$ denotes the operator norm. Then, with a probability at least $1 - \delta$,*

$$\max_{z \in \mathcal{Z}} \|z\|_{V_T^{-1}} \leq L/G_T, \text{ where } G_T = \sqrt{\lambda + T\lambda_{\min}(\Sigma_{\max}) - \sqrt{8 \sum_{s=1}^{T} C_s \log(d/\delta)}}.$$

**Proof sketch.** To derive the upper bound, we use various results related to the positive definite matrix (detailed in Fact 1 of the supplementary material). First, if $V_T$ is a positive definite matrix $V_T$, then for any $z \in \mathcal{Z}$, $\|z\|_{V_T^{-1}} \leq \|z\|_2 \sqrt{\lambda_{\max}(V_n^{-1})} = \|z\|_2 / \sqrt{\lambda_{\min}(V_n)}$. Thus, $\max_{z \in \mathcal{Z}} \|z\|_{V_T^{-1}} \leq \|z\|_2 / \sqrt{\lambda_{\min}(V_n)} \leq L / \sqrt{\lambda_{\min}(V_n)}$. Since $\{Z_s\}_{s=1}^{T}$ is a finite adapted sequence of self-adjoint matrices (i.e., $Z_s$ is $\mathcal{F}_s$-measurable for all $s$, where $\mathcal{F}_s$ represents all information available up to iteration $s$), we apply the Matrix Azuma inequality (Tropp, 2012) to get a high probability lower bound

on $\lambda_{\min}(V_n)$, specifically we have shown that $\lambda_{\min}(V_T) \geq T\lambda_{\min}(\Sigma_{\max}) - \sqrt{8\sum_{s=1}^{T} C_s \log{(d/\delta)}}$ holds with probability at least $1 - \delta$. Using this bound, we get the desired upper bound $L/G_T$. The full proof of Theorem 1, along with all other missing proofs, are provided in Section A.

This result shows that the upper bound can be expressed in terms of the number of adapted samples used to construct the matrix $V_T$, and it decays at a sub-linear rate as the number of samples $(T)$ increases. Notably, this result is of independent interest, as it provides valuable insights beyond the specific application of our work. Next, we give an upper bound on the worst sub-optimality gap in terms of the upper bound on the estimation error of the reward difference between any triplet consisting of a context and two arms.

**Lemma 1.** *Let $D_T = \{x_s, a_{s,1}, a_{s,2}, y_s\}_{s=1}^{T}$ be the preference dataset collected up to the iteration $T$ and $\hat{f}_T$ represent the estimate of latent reward function $f$ learned from $D_T$. With probability at least $1 - \delta, \forall c \in \mathcal{C}, a, b \in \mathcal{A}: \left| [f(\varphi(c, a)) - f(\varphi(c, b))] - \left[ \hat{f}_T(\varphi(c, a)) - \hat{f}_T(\varphi(c, b)) \right] \right| \leq \beta_T(c, a, b).$ If $a^\star = \arg\max_{a \in \mathcal{A}} f(\varphi(c, a))$ and $\pi(c)$ is the arm selected by policy for context $c$, then, with a probability at least $1 - \delta$, the worst sub-optimality gap for a policy that greedily selects an arm for a given context is upper bounded by: $\Delta_T^\pi \leq \max_{c \in \mathcal{C}} \beta_T(c, a^\star, \pi(c)).$*

The proof follows by starting with the worst sub-optimality gap definition in Eq. (1) and then applying a series of algebraic manipulations to derive the stated result. Our next results give an upper bound on $\beta_T(c, a, b)$ when $\texttt{Neural-ADB}$ uses different arm selection strategies.

**Lemma 2.** *Let $\nu_T = (\beta_T + B\sqrt{\lambda/\kappa_\mu} + 1)\sqrt{\kappa_\mu/\lambda}$, where $\beta_T = (1/\kappa_\mu)\sqrt{\tilde{d} + 2\log(1/\delta)}$ and $\delta \in (0, 1)$. If $w \geq poly(T, L, K, 1/\kappa_\mu, L_\mu, 1/\lambda_0, 1/\lambda, \log(1/\delta))$, then, with a probability of at least $1 - \delta$, for $\texttt{Neural-ADB}$ with (i) UCB-based arm selection strategy, for all $c \in \mathcal{C}: \beta_T(c, a, b) = \nu_T\sigma_T(c, a^\star, \pi(c)) + 2\varepsilon'_{w,T}$, (ii) TS-based arm selection strategy, for all $c \in \mathcal{C}: \beta_T(c, a, b) = \nu_T\log\left(KT^2\right)\sigma_T(c, a^\star, \pi(c)) + 2\varepsilon'_{w,T}$, where $K$ denotes the maximum number of arms available in each iteration, and $\varepsilon'_{w,T} = C_2 w^{-1/6}\sqrt{\log w}L^3 (T/\lambda)^{4/3}$ for some absolute constant $C_2 > 0$, is the approximation error that decreases as the width of the NN $(w)$ increases.*

Equipped with Theorem 1, Lemma 1, and Lemma 2, we will now provide an upper bound on the worse sub-optimality gap for a policy learned by $\texttt{Neural-ADB}$ while using UCB- and TS-based arm selection strategy for a given context.

**Theorem 2** (UCB). *Let the conditions in Theorem 1 and Lemma 2 hold. Then, with a probability with at least $1 - \delta$, the worst sub-optimality gap of $\texttt{Neural-ADB}$ when using UCB-based arm selection strategy is upper bounded by $\Delta_T^\pi \leq \left(\frac{\nu_T L}{G_T}\right)\sqrt{\frac{\lambda}{\kappa_\mu w}} + 2\varepsilon'_{w,T} = \tilde{O}\left(\sqrt{\frac{\tilde{d}}{T}}\right).$*

**Theorem 3** (TS). *Let the conditions in Theorem 1 and Lemma 2 hold. Then, with a probability with at least $1 - \delta$, the worst sub-optimality gap of $\texttt{Neural-ADB}$ when using TS-based arm selection strategy is upper bounded by $\Delta_T^\pi \leq \left(\frac{\nu_T L\log\left(KT^2\right)}{G_T}\right)\sqrt{\frac{\lambda}{\kappa_\mu w}} + 2\varepsilon'_{w,T} = \tilde{O}\left(\sqrt{\frac{\tilde{d}}{T}}\right).$*

The proof follows by applying Lemma 2, setting $z' = \varphi(c, a^\star) - \varphi(c, \pi(c))$ in Eq. (6), and then using Theorem 1. Note that $\varepsilon'_{w,T} = O(1/T)$ and $\tilde{d} = \tilde{o}(\sqrt{T})$ as long as the NN width $w$ is large enough (Zhou et al., 2020; Zhang et al., 2021; Verma et al., 2025). Above Theorem 2 and Theorem 3 show that the worst sub-optimality gap of the policy learned by $\texttt{Neural-ADB}$ with UCB- and TS-based arm selection strategies decreases at a sub-linear rate with respect to the size of preference dataset, specifically at rate of $\tilde{O}((\tilde{d}/T)^{\frac{1}{2}})$, where $\tilde{O}$ hides the logarithmic factors and constants. Further, the decay rate of the worst sub-optimality gap for $\texttt{Neural-ADB}$ improves by a factor of $\tilde{O}((\tilde{d}\log T)^{\frac{1}{2}})$ compared to exiting algorithms (Mehta et al., 2023; Das et al., 2024), thereby bridging the gap between theory and practice.

### 3.4 ACTIVE DUELING BANDITS WITH REGRET MINIMIZATION

We start by defining the *cumulative regret* (or 'regret' for brevity) of a policy. After receiving preference feedback for $T$ pairs of arms, the regret of a sequential arm selection

policy is given by: $\mathfrak{R}_T = \sum_{t=1}^{T} \left[ f(\varphi(c_t, a_t^\star)) - (f(\varphi(c_t, a_{t,1})) + f(\varphi(a_{t,2}))) / 2 \right]$, where $a_t^\star = \arg\max_{a \in \mathcal{A}_t} f(\varphi(c_t, a))$ denotes the arm that maximizes the reward function for a given context $c_t$.

In many real-world applications, such as medical treatment design (Lai and Robbins, 1985; Bengs et al., 2021) and content moderation (Avadhanula et al., 2022), both actively selecting arms and minimizing regret is required. For instance, in personalized medical treatment, active learning is used to identify the most informative treatments to test, while cumulative regret minimization ensures the system continually adapts to deliver better patient outcomes. Such scenarios also arise in other fields, such as dynamic pricing and personalized education, enabling systems to make smarter decisions, reduce suboptimal choices, and optimize overall performance as they gather more valuable data.

Since the arm selection strategies in `Neural-ADB` are directly adapted from UCB- and TS-based algorithms for contextual dueling bandits of Verma et al. (2025), the regret upper bounds for these algorithms also apply to `Neural-ADB`. For completeness, we state the regret upper bounds of `Neural-ADB` as follows.

**Corollary 1** (Regret Upper Bound). *(Verma et al., 2025, Theorem 2 and Theorem 3) Let $\lambda > \kappa_\mu$ and $w \geq poly(T, L, K, 1/\kappa_\mu, L_\mu, 1/\lambda_0, 1/\lambda, \log(1/\delta))$. Then, with a probability of at least $1 - \delta$, the regret of* `Neural-ADB` *when using UCB- or TS-based arm selection strategy is upper bounded by* $\mathfrak{R}_T = \widetilde{O}\left( \left( \frac{\sqrt{\widetilde{d}}}{\kappa_\mu} + \sqrt{\frac{\lambda}{\kappa_\mu}} \right) \sqrt{T\widetilde{d}} \right).$

Ignoring logarithmic factors and constants, the asymptotic growth rates of `Neural-ADB` with UCB- and TS-based arm selection strategy are identical and sub-linear.

## 4 EXPERIMENTS

To validate our theoretical results, we empirically evaluate the performance of our algorithms on different problem instances of synthetic datasets. Specifically, we use two commonly used synthetic functions adopted from existing works on neural bandits (Zhou et al., 2020; Zhang et al., 2021; Dai et al., 2023; Verma et al., 2025): $f(x) = 10(x^\top \theta)^2$ (Square) and $f(x) = 2\sin(x^\top \theta)$ (Sine). All experiments are repeated 10 times, and we report the average worst suboptimality gap with 95% confidence intervals (depicted as vertical lines on each curve).

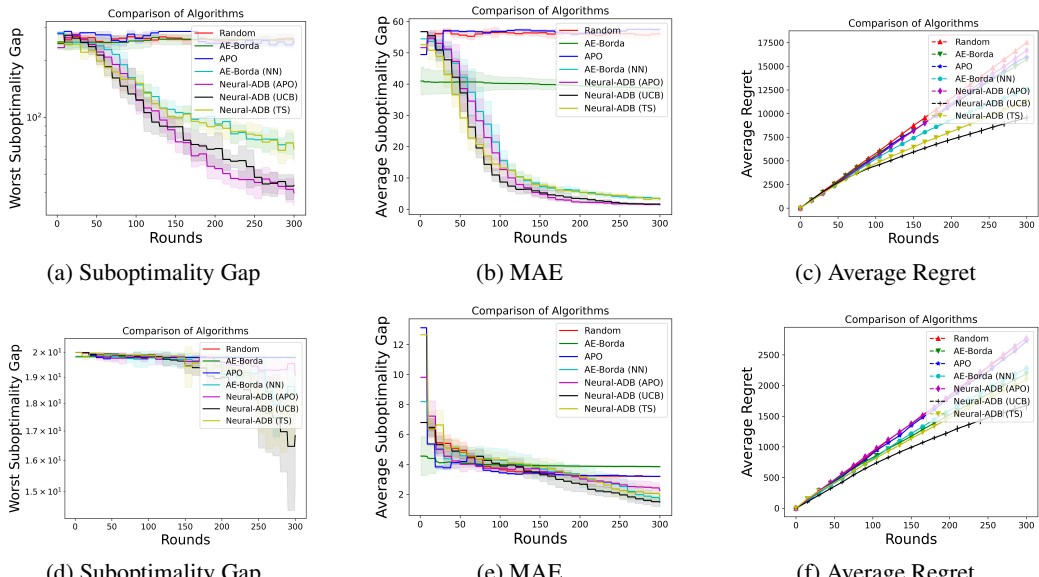

(a) Suboptimality Gap        (b) MAE        (c) Average Regret

(d) Suboptimality Gap        (e) MAE        (f) Average Regret

Figure 1: Performance comparison of `Neural-ADB` against different active dueling bandit algorithms on synthetic functions: Square function (top row) and Sine function (bottom row).

**Synthetic dataset.** We generate sample features for each context-arm pair in a $d$-dimensional space. Let $x_{t,a}$ be the context-arm feature vector for context $c_t$ and an arm $a$. For all $t \geq 1$, $x_{t,a}$ is sampled uniformly at random from $(-1, 1)^d$. We keep the number of arms constant across all rounds, denoted

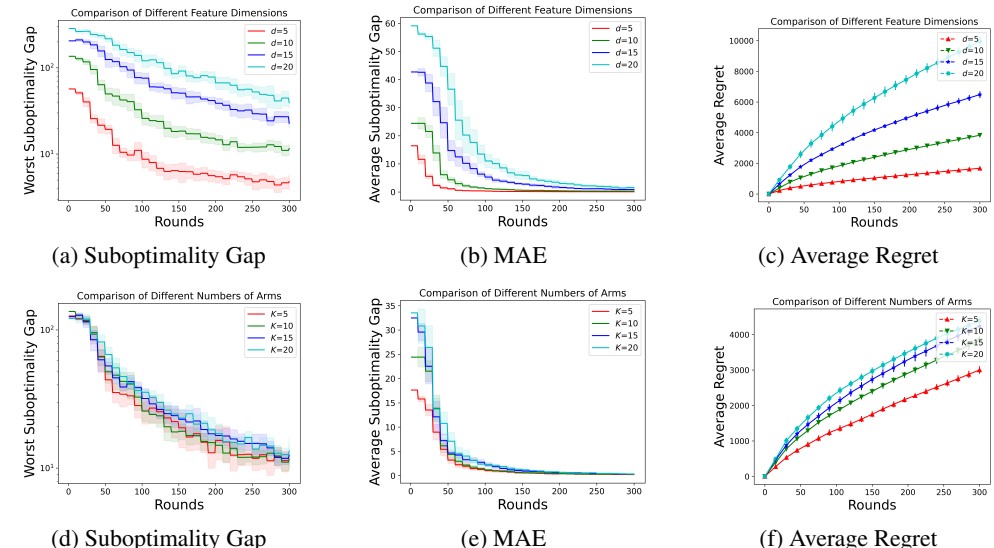

Figure 2: Performance of `Neural-ADB` (UCB) on the Square function, evaluated across varying input dimensions (top row) and numbers of arms (bottom row).

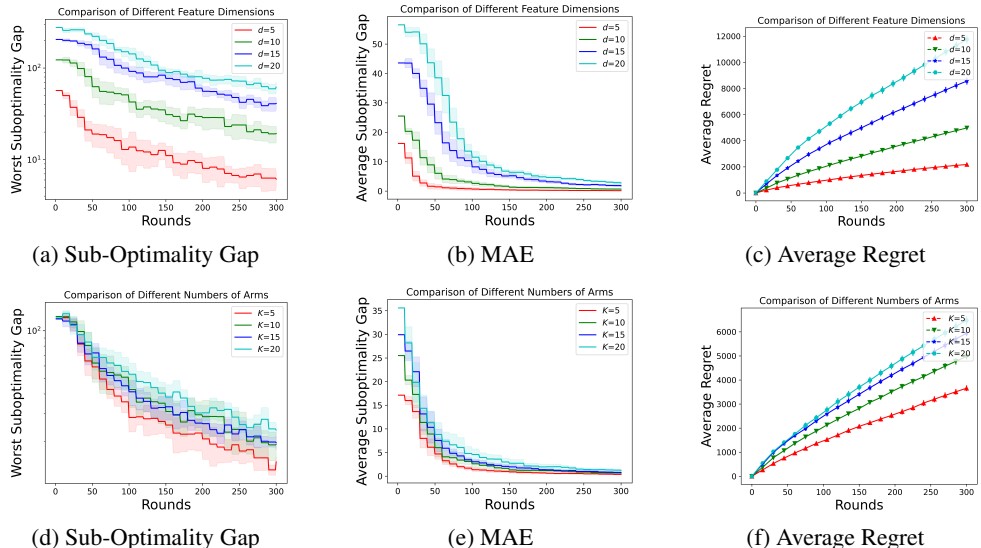

Figure 3: Performance of `Neural-ADB` (TS) on the Square function, evaluated across varying input dimensions (top row) and numbers of arms (bottom row).

by $K$. In our experiments, the binary preference feedback indicating whether $x_{t,1}$ preferred over $x_{t,2}$ (representing human preference feedback) is sampled from a Bernoulli distribution with parameter $\mu\left(f(x_{t,1}) - f(x_{t,2})\right)$, where $f$ is either a Square or Sine function..

**Reward function estimation.** We use a neural network with 2 hidden layers with width 50 to estimate the latent reward function, $\lambda = 1.0$, $\delta = 0.05$, $d = 20$, $K = 10$, $T = 1000$, and fixed value of $\nu_T = \nu = 1.0$ in all our experiments (unless we specifically indicate $d$ and $K$). Note that we did not perform any hyperparameter search for `Neural-ADB`, whose performance can be further improved by doing the hyperparameter search.

**Comparison with baselines.** We compare the worst suboptimality gap (defined in Eq. (1)), MAE (average suboptimality gap, i.e., $\sum_{t=1}^{T}\left[\max_{a \in \mathcal{A}} f(\varphi(c_t, a)) - f\left(\varphi(c_t, \pi(c))\right)\right]/T$, and average regret (defined in Section 3.4) against the different baselines of active contextual dueling bandits to evaluate the performance of UCB- and TS-variant of `Neural-ADB`. We use three baselines: Random, AE-Borda (Mehta et al., 2023), AE-DPO (Mehta et al., 2023), APO (Das et al., 2024),

and the neural variants of AE-Borda and APO in which we use a neural network to estimate the latent reward function. They are named AE-Borda (NN) and `Neural-ADB` (APO) respectively. Experimental results in Fig. 1 show that our algorithm, `Neural-ADB` (UCB), outperforms other baselines in almost all synthetic functions (i.e., square function and sine function) in terms of the suboptimality gap. Moreover, both UCB- and TS-variants of `Neural-ADB` also outperform other baselines on all synthetic functions in MAE and average regret. We have included more comparisons of our approach with other baselines in other settings (e.g., different $d$ or $K$) in Section C.

**Varying dimensions and arms vs. performance.** As we increase the dimension of the context-arm feature vectors ($d$) and number of arms ($K$), the problem becomes more challenging. To assess how the changes in $K$ and $d$ affect the performance of our proposed algorithms, we vary $K = \{5, 10, 15, 20\}$ and $d = \{5, 10, 15, 20\}$, while keeping the other parameters fixed. As expected, the performance of our algorithms gets worse with higher values of $K$ and $d$, as shown in Fig. 2. We have included similar results for `Neural-ADB` (TS) in Fig. 3.

## 5 RELATED WORK

This section reviews the most relevant work to our setting, i.e., contextual dueling bandits and active contextual dueling bandits. We discuss related topics, such as neural contextual bandits and finite-armed dueling bandits, in Section A.1.

**Contextual Dueling Bandits.** Many real-life applications, such as online recommendations, content moderation, medical treatment design, prompt optimization, and aligning large language models, can be effectively modeled using contextual dueling bandits, where a learner observes a context (additional information before selecting a pair of arms) and then selects the arms based on that context and observes preference feedback for the selected arms. Since the number of context-arm pairs can be potentially large or even infinite, the mean latent reward of each context-arm is assumed to be parameterized by an unknown function of its features. Common assumptions include linear reward models (Saha, 2021; Bengs et al., 2022; Di et al., 2023; Saha and Krishnamurthy, 2022; Li et al., 2024) and non-linear models (Verma et al., 2025). For our setting, we adopt the neural contextual dueling bandit algorithms proposed in (Verma et al., 2025) to construct confidence ellipsoids for the latent non-linear reward function. Note that `Neural-ADB` can incorporate alternative confidence ellipsoids by appropriately modifying Lemma 2. Furthermore, our work addresses an active learning problem and analyzes the convergence rate of the worst sub-optimality gap, whereas (Verma et al., 2025) focus on a regret minimization setting and derive upper bounds on cumulative regret.

**Active contextual dueling bandits.** The work most closely related to ours is active contextual dueling bandit (Mehta et al., 2023; Das et al., 2024), which takes a principled approach to actively collecting preference datasets. However, two key differences exist between our work and existing research: the non-linear reward function and the arm selection strategy. Existing studies typically assume a linear reward function, which may not be suitable for many real-world applications. Our work addresses this gap by extending the existing framework to incorporate non-linear reward functions in contextual dueling bandits. Additionally, existing approaches use different methods for selecting the pair of arms, leading to distinct arm selection strategies compared to ours. As a result of these differences in both the arm selection strategy and the non-linear reward function (which we estimate using a neural network), our analysis diverges significantly from that of prior work.

## 6 CONCLUSION

This paper studies the problem of active human preference feedback collection by modeling it as an active neural contextual dueling bandit problem. We propose `Neural-ADB`, a principled and practical algorithm designed for efficiently gathering human preference feedback in scenarios where the reward function is non-linear. Exploiting the neural contextual dueling bandit framework, `Neural-ADB` extends its applicability to a broad range of real-world applications, including online recommendation systems and LLM alignment. Our theoretical analysis demonstrates that the worst suboptimality gap of `Neural-ADB` decays at a sub-linear rate as the preference dataset grows. Finally, our experimental results further validate these theoretical findings. An interesting direction for future work is applying `Neural-ADB` to real-life applications such as LLM alignment. From a theoretical perspective, exploring the non-stationary setting is a promising future direction.

ETHICS STATEMENT

This work is primarily theoretical, focusing on the design and analysis of algorithms. The proposed methods do not directly involve human subjects, personal data, or real-world deployments. While the framework could potentially be applied in systems that interact with users, we emphasize that ethical considerations, such as fairness, privacy, and informed consent, must be addressed in practical deployments. Our primary goal is to advance the theoretical understanding of active preference data collection, and we do not anticipate any immediate negative societal impacts.

REPRODUCIBILITY STATEMENT

This paper primarily presents theoretical results, including formal proofs. All assumptions, definitions, and derivations are stated explicitly in the main text (see Section 3) and the Appendix. The details of our experimental setup are provided in Section 4 and the Appendix. Additionally, the code used in our experiments has been included in the supplementary material, enabling full reproduction of the results reported in this paper.

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

## A   APPENDIX

### A.1   ADDITIONAL RELATED WORK

**Neural Contextual Bandits.**   To model complex and non-linear reward functions, neural contextual bandits (Riquelme et al., 2018; Zhou et al., 2020; Zhang et al., 2021; Xu et al., 2022; Bae and Lee, 2025; Verma et al., 2025) use deep neural networks for reward function estimation. (Riquelme et al., 2018) employ multi-layer neural networks to learn arm embeddings and then use Thompson Sampling at the final layer for exploration. Zhou et al. (2020) propose the first neural contextual bandit algorithm with sub-linear regret guarantees, using a UCB exploration strategy. Building on this, Zhang et al. (2021) propose an algorithm with a TS exploration strategy. Ban et al. (2022) introduces an adaptive exploration strategy incorporating an auxiliary neural network to estimate the potential gain of the exploitation neural network, diverging from traditional UCB and TS exploration strategies. To reduce the computational overhead of using gradient-based features, Xu et al. (2022) only perform UCB-based exploration on the final layer of the neural network. More recent works (Bae and Lee, 2025; Verma et al., 2025) extend these techniques to handle neural contextual bandit settings with binary feedback (i.e., neural logistic bandits).

**Finite-Armed Dueling Bandits.**   Learning from preference feedback has been extensively studied in the bandit literature. In the finite-armed dueling bandits setting, the learner aims to find the best arm while only observing preference feedback for two selected arms (Yue and Joachims, 2009; 2011; Yue et al., 2012). To determine the best arm in dueling bandits, different criteria, such as the Borda winner, Condorcet winner, Copeland winner, or von Neumann winner, have been used while focusing on minimizing regret using only pairwise preference feedback (Ailon et al., 2014; Zoghi et al., 2014a;b; Gajane et al., 2015; Komiyama et al., 2015; Saha and Gopalan, 2018; 2019a;b; Verma et al., 2019; 2020a;b; Zhu et al., 2023). For a comprehensive overview of algorithms for various dueling bandits settings, we refer readers to the survey by Bengs et al. (2021).

## B   LEFTOVER PROOFS

To simplify the presentation, we use a common error probability of $\delta$ for all probabilistic statements. Our final results naturally follow by applying a union bound over all individual $\delta$. Next, we will describe the key properties of positive definite matrices crucial for the subsequent proofs. These properties form the basis for several key parts of our analysis.

**Fact 1** (Properties of a positive definite matrix). *Let $V_0 = \lambda \mathbb{I}_d$, $V_T = V_0 + \sum_{s=1}^{T} z_s z_s^\top$ be a positive definite matrix, where $\lambda > 0$, $z_s \in \mathbb{R}^d$, and $\{Z_s = z_s z_s^\top\}_{s=1}^T$ is a finite adapted sequence of self-adjoint matrices, i.e., $V_s$ and $Z_s$ are $\mathcal{F}_s$-measurable for all $s$, where $\mathcal{F}_s$ represents all information available up to $s$. We use $\lambda_{\max}(V_T)$ and $\lambda_{\min}(V_T)$ to denote the maximum and minimum eigenvalue of matrix $V_T$. Then, the following properties hold for $V_T$:*

*1. Let $\delta \in (0, 1)$, $\forall s \leq T: \|V_s - V_{s-1}\|^2 \leq C_s$, where $\|A\|$ denotes the operator norm. Then, using Theorem 7.1 and Corollary 7.2 of Tropp (2012), with probability at least $1 - \delta$,*

$$\mathbb{P}\left\{\lambda_{\max}\left(V_T - \mathbb{E}\left[V_T\right]\right) \geq \sqrt{8 \sum_{s=1}^{T} C_s \log\left(\frac{d}{\delta}\right)}\right\} \leq \delta.$$

*2. $\lambda_{\max}(V_T) = -\lambda_{\min}(-V_T)$.*

*3. Let $\lambda_i(V)$ be the $i$-th eigenvalue of matrix $V$. If $W$ is any Hermitian matrix, then, from Weyl's inequality:*

*1. $\lambda_i(V_T) + \lambda_{\min}(W) \leq \lambda_i(V_T + W) \leq \lambda_i(V_T) + \lambda_{\max}(W)$ and*

$$2.\ \lambda_i(V_T) - \lambda_{\max}(W) \leq \lambda_i(V_T - W) \leq \lambda_i(V_T) - \lambda_{\min}(W).$$

4. *Let* $\forall z \in \mathbb{R}^d:\ \|z\|_2 \leq L.$ *Then,* $\max_{z \in \mathbb{R}^d} \|z\|_{V_T^{-1}} \leq \|z\|_2 \sqrt{\lambda_{\max}(V_T^{-1})} \leq L/\sqrt{\lambda_{\min}(V_T)}.$

5. *For* $a > 0:\ \|az\|_{V_T} = a\|z\|_{V_T}$ *and* $\lambda_i(aV_T) = a\lambda_i(V_T).$

## B.1 PROOF OF THEOREM 1

We now prove the upper bound on the maximum Mahalanobis norm of a vector from the fixed input space, measured with respect to the inverse of a positive definite Gram matrix defined by finite, adapted samples from the same input space.

**Theorem 1.** *Let* $\{Z_s = z_s z_s^\top\}_{s=1}^T$ *be a finite adapted sequence of self-adjoint matrices in* $\mathbb{R}^d$. *Define* $\mathbb{E}\left[z_s z_s^\top\right] = \Sigma_s \leq \Sigma_{\max}$, $V_0 = \lambda\mathbb{I}_d$, $V_T = \lambda\mathbb{I}_d + \sum_{s=1}^T z_s z_s^\top$. *Assume* $\|z_s\|_2 \leq L$ *for all* $z \in \mathcal{Z} \subset \mathbb{R}^d$, $\lambda_{\min}(A)$ *denote the minimum eigenvalue of a matrix* $A$, *and* $\forall s \leq T :$ $\|V_s - V_{s-1}\|^2 \leq C_s$, *where* $\|V\|$ *denotes the operator norm. Then, with a probability at least* $1 - \delta$, $\max_{z \in \mathcal{Z}} \|z\|_{V_T^{-1}} \leq L/G_T$, *where* $G_T = \sqrt{\lambda + T\lambda_{\min}(\Sigma_{\max}) - \sqrt{8\sum_{s=1}^T C_s \log(d/\delta)}}.$

*Proof.* Using Property 1 in Fact 1 with $Y_T - \mathbb{E}[Y_T] = \mathbb{E}[V_T] - V_T$, we have

$$\mathbb{P}\left\{\lambda_{\max}\left(\mathbb{E}[V_T] - V_T\right) \geq \tau\right\} \leq d\exp\left(\frac{-\tau^2}{8\sum_{s=1}^T C_s}\right)$$

$$\implies \mathbb{P}\left\{-\lambda_{\min}\left(-(\mathbb{E}[V_T] - V_T)\right) \geq \tau\right\} \leq d\exp\left(\frac{-\tau^2}{8\sum_{s=1}^T C_s}\right) \quad \text{(Property 2 in Fact 1)}$$

$$\implies \mathbb{P}\left\{\lambda_{\min}\left(V_T - \mathbb{E}[V_T]\right) \leq -\tau\right\} \leq d\exp\left(\frac{-\tau^2}{8\sum_{s=1}^T C_s}\right).$$

Using upper bound on $\lambda_{\min}(V_T - \mathbb{E}[V_T])$ from Property 3 in Fact 1, we get

$$\implies \mathbb{P}\left\{\lambda_{\min}(V_T) - \lambda_{\min}(\mathbb{E}[V_T]) \leq -\tau\right\} \leq d\exp\left(\frac{-\tau^2}{8\sum_{s=1}^T C_s}\right)$$

$$\implies \mathbb{P}\left\{\lambda_{\min}(V_T) \leq \lambda_{\min}(\mathbb{E}[V_T]) - \tau\right\} \leq d\exp\left(\frac{-\tau^2}{8\sum_{s=1}^T C_s}\right).$$

Note that $\mathbb{E}[V_T] = \mathbb{E}\left[\lambda\mathbb{I}_d + \sum_{t=1}^T z_s z_s^\top\right] = \lambda\mathbb{I}_d + \sum_{t=1}^T \mathbb{E}\left[\lambda\mathbb{I}_d z_s z_s^\top\right] = \lambda\mathbb{I}_d + \sum_{t=1}^T \Sigma_s \leq \lambda\mathbb{I}_d + T\Sigma_{\max}$. Thus, we get

$$\mathbb{P}\left\{\lambda_{\min}(V_T) \leq \lambda + T\lambda_{\min}(\Sigma_{\max}) - \sqrt{8\sum_{s=1}^T C_s \log\left(\frac{d}{\delta}\right)}\right\} \leq \delta.$$

Therefore, with probability at least $1 - \delta$, $\lambda_{\min}(V_T) \geq \lambda + T\lambda_{\min}(\Sigma_{\max}) - \sqrt{8\sum_{s=1}^T C_s \log(d/\delta)}$. Using Property 4 in Fact 1, we now use to prove our key result as follows:

$$\max_{z \in \mathcal{Z}} \|z\|_{V_T^{-1}} \leq L/\sqrt{\lambda_{\min}(V_T)}$$

$$\leq L/\sqrt{\lambda + T\lambda_{\min}(\Sigma_{\max}) - \sqrt{8\sum_{s=1}^T C_s \log\left(\frac{d}{\delta}\right)}}$$

$$= L/G_T$$

$$\implies \max_{z \in \mathcal{Z}} \|z\|_{V_T^{-1}} \leq= L/G_T. \qquad \square$$

B.2 PROOF OF LEMMA 1 AND LEMMA 2

Our next results gives an upper bound of worst sub-optimality gap in terms of the upper bound of estimation error in the reward difference between any triple of context and two arms.

**Lemma 1.** *Let $D_T = \{x_s, a_{s,1}, a_{s,2}, y_s\}_{s=1}^T$ be the preference dataset collected up to the iteration $T$ and $\hat{f}_T$ represent the estimate of latent reward function $f$ learned from $D_T$. With probability at least $1 - \delta, \forall c \in \mathcal{C}, \ a, b \in \mathcal{A}: \left| [f(\varphi(c,a)) - f(\varphi(c,b))] - \left[ \hat{f}_T(\varphi(c,a)) - \hat{f}_T(\varphi(c,b)) \right] \right| \leq \beta_T(c,a,b)$. If $a^\star = \operatorname{argmax}_{a \in \mathcal{A}} f(\varphi(c,a))$ and $\pi(c)$ is the arm selected by policy for context $c$, then, with a probability at least $1 - \delta$, the worst sub-optimality gap for a policy that greedily selects an arm for a given context is upper bounded by: $\Delta_T^\pi \leq \max_{c \in \mathcal{C}} \beta_T(c, a^\star, \pi(c))$.*

*Proof.* Define $a^\star = \operatorname{argmax}_{a \in \mathcal{A}} f(\varphi(c,a))$. Recall the definition of worst suboptimality across all contexts, which is :

$$\Delta_{\mathcal{D}_T}^\pi = \max_{c \in \mathcal{C}} \left[ \max_{a \in \mathcal{A}} f(\varphi(c,a)) - f(\varphi(c, \pi(c))) \right]$$

$$= \max_{c \in \mathcal{C}} \left[ f(\varphi(c, a^\star)) - f(\varphi(c, \pi(c))) \right]$$

$$= \max_{c \in \mathcal{C}} \left[ f(\varphi(c, a^\star)) - f(\varphi(c, \pi(c))) + \hat{f}_T(\varphi(c, a^\star)) - \hat{f}_T(\varphi(c, a^\star)) \right]$$

$$\leq \max_{c \in \mathcal{C}} \left| [f(\varphi(c, a^\star)) - f(\varphi(c, \pi(c)))] + \left[ \hat{f}_T(\varphi(c, \pi(c))) - \hat{f}_T(\varphi(c, a^\star)) \right] \right|$$

$$= \max_{c \in \mathcal{C}} \left| [f(\varphi(c, a^\star)) - f(\varphi(c, \pi(c)))] - \left[ \hat{f}_T(\varphi(c, a^\star)) - \hat{f}_T(\varphi(c, \pi(c))) \right] \right|$$

$$\implies \Delta_{\mathcal{D}_T}^\pi \leq \max_{c \in \mathcal{C}} \beta_T(c, a^\star, \pi(c)).$$

The inequality follows from the fact we have greedy policy, i.e., $\pi(c) = \operatorname{argmin}_{a \in \mathcal{A}} \hat{f}_T(\varphi(c,a))$ for any context $c$. Therefore, if $\pi(c) \neq a^\star$, then $\hat{f}_T(\varphi(c, \pi(c))) \geq \hat{f}_T(\varphi(c, a^\star))$ must hold. □

**Lemma 2.** *Let $\nu_T = (\beta_T + B\sqrt{\lambda/\kappa_\mu} + 1)\sqrt{\kappa_\mu/\lambda}$, where $\beta_T = (1/\kappa_\mu)\sqrt{\tilde{d} + 2\log(1/\delta)}$ and $\delta \in (0,1)$. If $w \geq poly(T, L, K, 1/\kappa_\mu, L_\mu, 1/\lambda_0, 1/\lambda, \log(1/\delta))$, then, with a probability of at least $1 - \delta$, for $\texttt{Neural-ADB}$ with (i) UCB-based arm selection strategy, for all $c \in \mathcal{C} : \beta_T(c,a,b) = \nu_T \sigma_T(c, a^\star, \pi(c)) + 2\varepsilon'_{w,T}$, (ii) TS-based arm selection strategy, for all $c \in \mathcal{C} : \beta_T(c,a,b) = \nu_T \log\left(KT^2\right) \sigma_T(c, a^\star, \pi(c)) + 2\varepsilon'_{w,T}$, where $K$ denotes the maximum number of arms available in each iteration, and $\varepsilon'_{w,T} = C_2 w^{-1/6}\sqrt{\log w} L^3 (T/\lambda)^{4/3}$ for some absolute constant $C_2 > 0$, is the approximation error that decreases as the width of the NN $(w)$ increases.*

*Proof.* Recall that we are using the arm-selection strategies proposed in (Verma et al., 2025). Since their confidence bounds hold for any adapted sequence of contexts, the proof of the first part follows directly from Theorem 1 in (Verma et al., 2025), while the second part follows from Lemma 10 together with Eq. (27) of (Verma et al., 2025). □

**Remark 1.** *We adopt the arm selection strategies from the existing neural dueling bandit algorithms in (Verma et al., 2025), which assume $\tilde{d} = o(T)$. In some cases, $\tilde{d} = \Omega(T)$ (Ban et al., 2022; Deb et al., 2024), which may result in a constant convergence rate. However, our objective is to demonstrate the use of the neural network for estimating non-linear reward functions in active contextual dueling bandits. Since neural dueling bandit algorithms primarily influence the arm selection strategy, we can incorporate any variants of these algorithms by making appropriate modifications to Lemma 2.*

B.3 PROOF OF THEOREM 2 AND THEOREM 3

Equipped with Theorem 1, Lemma 1, and Lemma 2, we will next prove the upper bound on the worst sub-optimality gap for a policy learned by $\texttt{Neural-ADB}$ while using UCB- and TS-based arm selection strategy for a given context.

**Theorem 2** (UCB). *Let the conditions in Theorem 1 and Lemma 2 hold. Then, with a probability with at least $1 - \delta$, the worst sub-optimality gap of* `Neural-ADB` *when using UCB-based arm selection strategy is upper bounded by* $\Delta_T^\pi \leq \left(\frac{\nu_T L}{G_T}\right)\sqrt{\frac{\lambda}{\kappa_\mu w}} + 2\varepsilon_{w,T}' = \tilde{O}\left(\sqrt{\frac{\tilde{d}}{T}}\right).$

*Proof.* Using Lemma 2 and setting value of $\beta_T(c, a^\star, \pi(c))$ using Lemma 2 and Eq. (6), we have

$$\Delta_T^\pi \leq \max_{c \in \mathcal{C}} \beta_T(c, a^\star, \pi(c)) \qquad \text{(from Lemma 1)}$$

$$\leq \max_{c \in \mathcal{C}} \left(\nu_T \sigma_T(c, a^\star, \pi(c)) + 2\varepsilon_{w,T}'\right). \qquad \text{(from Lemma 2)}$$

As $\nu_T$ and $\varepsilon_{w,T}'$ independent of context $c$, we get

$$\Delta_T^\pi \leq \nu_T \max_{c \in \mathcal{C}} \left(\sigma_T(c, a^\star, \pi(c))\right) + 2\varepsilon_{w,T}'$$

$$= \nu_T \max_{c \in \mathcal{C}} \left(\sqrt{\frac{\lambda}{\kappa_\mu}} \left\|\frac{\varphi(c, a^\star) - \varphi(c, \pi(c))}{\sqrt{w}}\right\|_{V_T^{-1}}\right) + 2\varepsilon_{w,T}' \qquad \text{(using Eq. (6))}$$

$$= \nu_T \max_{c \in \mathcal{C}} \left(\sqrt{\frac{\lambda}{\kappa_\mu w}} \left\|\varphi(c, a^\star) - \varphi(c, \pi(c))\right\|_{V_T^{-1}}\right) + 2\varepsilon_{w,T}' \qquad \text{(Property 5 in Fact 1)}$$

$$= \nu_T \sqrt{\frac{\lambda}{\kappa_\mu w}} \max_{c \in \mathcal{C}} \left(\left\|\varphi(c, a^\star) - \varphi(c, \pi(c))\right\|_{V_T^{-1}}\right) + 2\varepsilon_{w,T}'$$

$$\leq \nu_T \sqrt{\frac{\lambda}{\kappa_\mu w}} \left(\frac{L}{\sqrt{\lambda + T\lambda_{\min}(\Sigma_{\max})} - \sqrt{8\sum_{s=1}^{T} C_s \log\left(\frac{d}{\delta}\right)}}\right) + 2\varepsilon_{w,T}' \qquad \text{(using Theorem 1)}$$

$$\leq \tilde{O}\left(\sqrt{\frac{\tilde{d}}{T}}\right). \qquad \qed$$

**Theorem 3** (TS). *Let the conditions in Theorem 1 and Lemma 2 hold. Then, with a probability with at least $1 - \delta$, the worst sub-optimality gap of* `Neural-ADB` *when using TS-based arm selection strategy is upper bounded by* $\Delta_T^\pi \leq \left(\frac{\nu_T L \log(KT^2)}{G_T}\right)\sqrt{\frac{\lambda}{\kappa_\mu w}} + 2\varepsilon_{w,T}' = \tilde{O}\left(\sqrt{\frac{\tilde{d}}{T}}\right).$

*Proof.* Using Lemma 2 and setting value of $\beta_T(c, a^\star, \pi(c))$ using Lemma 2 and Eq. (6), we have

$$\Delta_T^\pi \leq \max_{c \in \mathcal{C}} \beta_T(c, a^\star, \pi(c)) \qquad \text{(from Lemma 1)}$$

$$\leq \max_{c \in \mathcal{C}} \left(\nu_T \log\left(KT^2\right) \sigma_T(c, a^\star, \pi(c)) + 2\varepsilon_{w,T}'\right). \qquad \text{(from Lemma 2)}$$

The value of $\nu_T$ and $\varepsilon_{w,T}'$ independent of context $c$. By following similar steps to those in the proof of Theorem 2, we have

$$\Delta_T^\pi \leq \nu_T \log\left(KT^2\right) \max_{c \in \mathcal{C}} \left(\sigma_T(c, a^\star, \pi(c))\right) + 2\varepsilon_{w,T}'$$

$$= \nu_T \log\left(KT^2\right) \max_{c \in \mathcal{C}} \left(\sqrt{\frac{\lambda}{\kappa_\mu}} \left\|\frac{\varphi(c, a^\star) - \varphi(c, \pi(c))}{\sqrt{w}}\right\|_{V_T^{-1}}\right) + 2\varepsilon_{w,T}'$$

$$= \nu_T \log\left(KT^2\right) \max_{c \in \mathcal{C}} \left(\sqrt{\frac{\lambda}{\kappa_\mu w}} \left\|\varphi(c, a^\star) - \varphi(c, \pi(c))\right\|_{V_T^{-1}}\right) + 2\varepsilon_{w,T}'$$

$$= \nu_T \log\left(KT^2\right) \sqrt{\frac{\lambda}{\kappa_\mu w}} \max_{c \in \mathcal{C}} \left(\left\|\varphi(c, a^\star) - \varphi(c, \pi(c))\right\|_{V_T^{-1}}\right) + 2\varepsilon_{w,T}'$$

$$\leq \nu_T \log\left(KT^2\right) \sqrt{\frac{\lambda}{\kappa_\mu w}} \left(\frac{L}{\sqrt{\lambda + T\lambda_{\min}(\Sigma_{\max})} - \sqrt{8\sum_{s=1}^{T} C_s \log\left(\frac{d}{\delta}\right)}}\right) + 2\varepsilon'_{w,T}$$

$$\leq \tilde{O}\left(\sqrt{\frac{\tilde{d}}{T}}\right). \qquad \square$$

## C  ADDITIONAL EXPERIMENTAL DETAILS AND RESULTS

### C.1  EXPERIMENTAL DETAILS

**Computational resources used for experiments.**  All experiments were conducted on a server equipped with an AMD EPYC 7543 32-Core Processor, 256GB of RAM, and 8 NVIDIA GeForce RTX 3080 GPUs.

**Practical considerations.**  Based on the neural tangent kernel (NTK) theory (Jacot et al., 2018), the initial gradient $g(x; \theta_0)$ can be used as the original feature vector $x$ as $g(x; \theta_0)$ effectively represents the random Fourier features of the NTK. To make our algorithm more practical, we use common practices in neural bandits (Zhou et al., 2020; Zhang et al., 2021; Verma et al., 2025). Specifically, we replaced the theoretical regularization parameter $\frac{1}{2} w\lambda \|\theta - \theta_0\|_2^2$ (where $w$ is the NN's width) with the simpler $\lambda \|\theta\|_2^2$ in the loss function (defined in Eq. (2)) that is used to train our NN. We retrain the neural network after every 20 rounds for 50 gradient steps across all experiments.

### C.2  ADDITIONAL EXPERIMENTAL RESULTS

Next, we present the additional experiment results comparing the performance of `Neural-ADB` varying input dimension $d$ (Fig. 4) and different numbers of arms $K$ (Fig. 5).

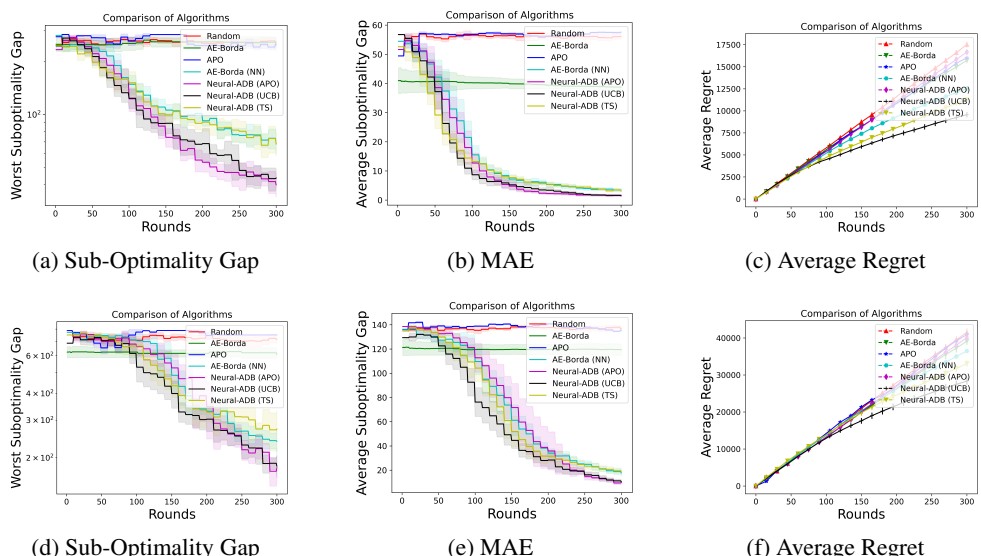

|     |     |     |
|-----|-----|-----|
| (a) Sub-Optimality Gap | (b) MAE | (c) Average Regret |
| (d) Sub-Optimality Gap | (e) MAE | (f) Average Regret |

Figure 4:  Performance comparison across different input dimensions $d$: $d = 20$ (first row) and $d = 40$ (second row). We set the number of arms to 10 and use the Square function for all experiments.

**Performance vs. neural network size.**  To investigate how performance varies with different neural network (NN) sizes, we used the Square and Cosine functions defined in the paper. We varied either the number of layers (with width = 32) or the width of the NN (with 2 layers), while keeping all other variables consistent with those in the paper. As shown in Fig. 6, we observed that selecting the appropriate size of NN is crucial for the given problem. Using a large NN for a simple problem leads

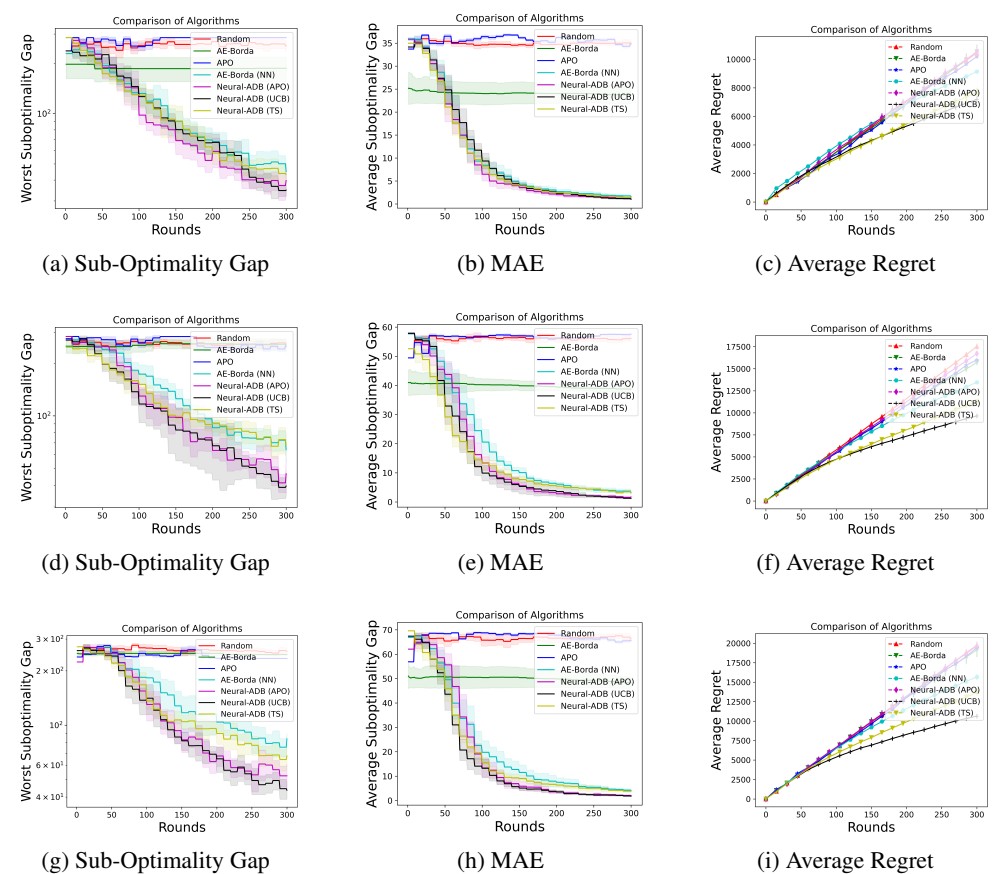

Figure 5: Performance comparison across different numbers of arms $K$: $K = 5$ (top row), $K = 10$ (middle row), and $K = 15$ (bottom row). We set the input dimension to 20 and use the Square function for all experiments.

to poor performance due to high bias in the estimation, while a smaller NN may not accurately be able to estimate the complex non-linear function.

### C.3 COMPUTATIONAL EFFICIENCY.

To discuss the computational efficiency of `Neural-ADB`, we follow the approach of (Verma et al., 2025) and consider the following two key aspects: size of the neural network and then the number of contexts and arms.

**Size of the neural network.** The primary computational cost in `Neural-ADB` arises from the neural network (NN) used to approximate the latent non-linear reward function. Given a context-arm feature vector of dimension $d$, an NN with $D$ hidden layers and $w$ neurons per layer incurs an inference cost of $O(dw + Dw^2 + w)$ per context-arm pair. The total number of parameters in the NN is $p = dw + Dw^2 + w$, and the training time per iteration is $O(\mathcal{E}\mathcal{P}Dw^2)$, where $\mathcal{E}$ is the number of training epochs and $\mathcal{P}$ is the number of observed context-arm pairs. Choosing an appropriate NN size is critical, as NNs that are too small may fail to accurately approximate the underlying non-linear reward function, while excessively large NNs can result in substantial training and inference overhead.

**Number of contexts and arms.** Let $K$ denote the number of arms and $p$ the total number of NN parameters. Since `Neural-ADB` uses NN gradients as context-arm features, the cost of computing gradients for all arms per context is $O(K^2 dp)$, where $d$ is the dimension of the context-arm feature vector. The cost of computing reward estimates and confidence terms for all context-arm pairs is $O(K^2 p)$ and $O(K^2 p^2)$, respectively. For arm selection, the first selection step requires $O(Kp + K)$,

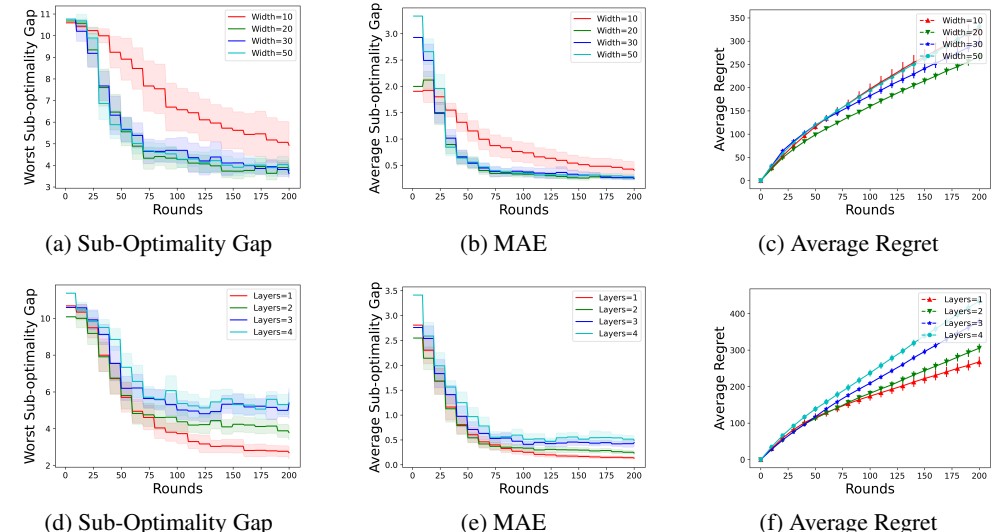

Figure 6: We compare performance across different neural network widths (first row) and numbers of hidden layers (second row), using the Square function in all experiments. All other parameters are kept fixed, except that the width is set to 32 when varying the number of layers.

consisting of reward estimation for all arms ($O(Kp)$) and then identifying the arm with the highest estimated reward ($O(K)$). The second arm selection incurs a cost of $O(Kp + (K-1)p^2)$, including reward estimation $O(Kp)$ and confidence term computation $O((K-1)p^2)$ relative to the first selected arm. Thus, the total computational cost for selecting a pair of arms per context is $O(K^2dp + K^2p^2)$. Since each context-arm pair is independent, gradients, reward estimates, and optimistic terms can be computed in parallel, reducing the overall cost to $O(dp+p^2)$ for each iteration.

