# OpenReview forum: "Provably Sample-Efficient Active Preference Data Collection"
_ICLR.cc/2026/Conference — Submitted to ICLR 2026_

### Official Review · Reviewer_iYkc · 2025-10-28

**Soundness:** 2
**Presentation:** 2
**Contribution:** 2
**Rating:** 2
**Confidence:** 4

**Summary:**

This paper studies active learning setting for non-linear reward/preferences. They propose Neural-ADB, an algorithm for efficiently collecting human preference feedback when the underlying reward function is non-linear, thereby moving beyond the linear assumption of existing active contextual/dueling bandit methods. The algorithm uses neural networks to estimate the unknown reward function from preference data following the Bradley-Terry-Luce model, combining context selection (adapted from Das et al. 2024) with UCB- or Thompson sampling-based arm selection strategies. The theoretical analysis introduces an upper bound (which I did not find novel, more on this later) on the maximum Mahalanobis norm for vectors from a fixed input space with respect to adapted Gram matrices (Theorem 1). The worst suboptimality gap of Neural-ADB decreases at $O(\sqrt{(\tilde{d}/T)})$, improving upon existing algorithms by a factor of $O(\sqrt{(\tilde{d} \log T)})$. The standard trade-off between exploration and exploitation is also balanced by context selection which maximizes the Mahalanobis norm difference between arm pairs to encourage diversity (exploration), while arm selection balances exploration through confidence bounds. The first arm is selected greedily to maximize estimated reward, while the second arm maximizes either UCB values or Thompson samples to ensure exploration. Experiments on synthetic functions (Square and Sine) with varying dimensions and numbers of arms demonstrate Neural-ADB's superior performance over baselines, including Random, AE-Borda, AE-DPO, and APO. The algorithm addresses real-world applications like online recommendations and LLM alignment where linear reward assumptions fail.

**Strengths:**

1. The paper combines theory and practical application by introducing neural networks for non-linear reward function estimation in active contextual dueling bandits. This addresses the limitation of existing methods that assume linearity. Their convergence rate of  $O(\sqrt{(\tilde{d}/T)})$ improves upon $O(\sqrt{(\tilde{d} \log T)})$ for existing methods.
2. The experiments in the paper evaluate the performance of their algorithm across multiple dimensions (suboptimality gap, MAE, average regret) and conditions (varying d, K, network architectures). They show consistent improvements over baselines and also provide practical insights about hyperparameter selection.

**Weaknesses:**

1. The paper suffers seriously from some over-claim. I did not find much difference between their approach and kernel bandits.
2. This is mainly a theoretical paper, and so the theoretical claims should be given more importance. As such, I did not find any novelty in their proof techniques.
3. Some references are missing. This is a crowded/well-studied space, and several baselines are also missing.

**Questions:**

1. It is not clear to me how the paper distinguishes itself from kernel bandits https://proceedings.mlr.press/v70/chowdhury17a/chowdhury17a-supp.pdf, https://proceedings.neurips.cc/paper/2021/file/6084e82a08cb979cf75ae28aed37ecd4-Paper.pdf, and what novelty lies in the algorithm. The exploration term and manipulations with the Mahalanobis norm are very similar to existing work. The only slight difference is the introduction of the duelling (BTL model). More clarification on this will be very helpful.
2. Lemma 2, Theorem 2, and Theorem 3 are standard applications of bandit proof techniques. A discussion on their novelty will be helpful.
3. More discussion on how this work is different than Neural contextual bandits, and how the NTK solution-based approaches are linked to the proof must be discussed.
4. The  BTL method is well studied; the generalized Plackett-Luce method, which covers BTL, should also be discussed. You already cite Principled reinforcement learning with human feedback from pairwise or k-wise comparisons, please compare against that. Possibly compare against Optimal Design for Human Preference Elicitation, which actually tests in the natural language domain.

---

### Official Review · Reviewer_7RY1 · 2025-10-28

**Soundness:** 2
**Presentation:** 2
**Contribution:** 2
**Rating:** 2
**Confidence:** 4

**Summary:**

The paper addresses the problem of actively collecting pairwise preference data in contextual settings where the underlying reward function can be nonlinear in the context–arm features. The proposed method, Neural-ADB, combines a neural preference model with an active data selection scheme. It trains a ReLU network on preference pairs, selects informative contexts using a discrepancy criterion based on initialization gradients, and chooses arms through either UCB or Thompson sampling strategies adapted from neural dueling bandits. The theoretical analysis, built under the Bradley–Terry–Luce (BTL) preference model and standard smoothness assumptions on the link function, claims that the worst-case sub-optimality gap of the learned policy scales as $\tilde{O}(\tilde{d}/T)$ where $\tilde{d}$ is an “effective dimension”. This paper also gives a new matrix concentration bound (Theorem 1). Experiments on synthetic non-linear rewards (square and sine) show improved sub-optimality gap, mean absolute error, and regret vs. linear baselines and simple neuralized variants.

**Strengths:**

1. The paper is well-motivated: it tackles a meaningful limitation in prior active contextual dueling bandit work, which typically assumes a linear reward structure. Extending the framework to nonlinear function classes through neural network surrogates is a natural and worthwhile direction, and the design remains explicitly exploration-aware.
2. The algorithmic decomposition is clear: Context selection (Eq. 3) + arm selection (UCB/TS) is conceptually clean, although it would be better if the paper briefly explains the behind ideas of these steps.
3.  This paper provides empirical sanity checks: Synthetic experiments systematically vary dimension $d$ and number of arms $K$, and report sub-optimality gap, MAE, and regret.

**Weaknesses:**

1. The proof of Theorem 1 (and thus the theorem as stated) is not correct. Specifically, the step after line 741 does not hold. $\lambda_{\min} (\mathbb{E}[V_{T}])$ is upper bounded by the right hand side of the equation inside $\mathbb{P}$, and thus this probability is not necessarily less than $\delta$. Perhaps in the statement of Theorem 1, A more sensible statement would assume a lower PSD bound $\Sigma_{\min}$ for $\Sigma_{s}$. In its current form, the theorem is hard to interpret, especially because the $\Sigma_{\max}$ could be arbitrarily large and hence the upper bound could be very small.
2. $C_{s}$ in Theorem 1 appears as a black-box term, yet the main result of this paper relies on it. It would be better to unpack what $C_s$ actually measures, and give some intuition when $G_{T}$ is greater than zero (if the lower bound $\Sigma_{\min}$ works, as discussed in the above bullet point), so that the bound makes sense.
3. Since the contribution is nonlinear active preference learning, it would help to position the result more directly against known linear/dueling or linear-contextual results (rates, assumptions, noise models). Even a short comparison table or paragraph would make it clearer what is genuinely new here versus what is being carried over from the linear literature.

**Questions:**

1. Could you please provide concrete bounds/estimates of $\tilde{d}$ and $\kappa_{\mu}$ for the synthetic distributions you use, and discuss regimes where it may scale poorly?
2. Could you please comment whether $p$ (dimension of the NN parameters) plays a role here?

---

Minors:
1. line 272: the proof is provides in Section B (not A).
2. line 266: Proof sketch of Theorem 1 has some notation issues. It uses $V_{n}$. I think it should be $V_{T}$, right?
3. It is not obvious why $\beta_{T}$ depends on $T$. I think it would be better if we make it clear when we first introduce it.
4. line 163: missed a ;.
5. line 127: there is an extra dot above $\mu$.
6. line 441: missed a space before $d$.

---

### Official Review · Reviewer_8VFo · 2025-10-29

**Soundness:** 2
**Presentation:** 2
**Contribution:** 2
**Rating:** 4
**Confidence:** 2

**Summary:**

As collecting human preference data can be expansive, this paper considers the problem of efficient strategies to collect human preference. They propose a neural contextual dueling bandit framework with non-linear latent reward function. They propose UCB-type arm selection and Thompson  sampling-type arm selection strategies and give theoretical upper bounds on their sub-optimality gaps. They also provide empirical evaluations over synthetic dataset.

**Strengths:**

I am not an expert on bandits. So please take my comments with less weight.

- Originality: use active dueling bandit framework for efficient preference dataset collection sounds reasonable and original.
- Clarity: Overall the paper is articulated and presented in a clear way.
- Quality: The overall quality look alright.

**Weaknesses:**

I am not an expert on bandits. So please take my comments with less weight.

- The theoretical result (Line 317) is said to improve previous results (Das et al., 2024) by a factor of $\tilde{O}(\tilde{d}\log T)$. While  (Das et al., 2024)  does have an additional $\log T$, it does not seem to have $\tilde{d}$ which may be a function of $T$. Could you clarify it?
- Methods are evaluated on synthetic data only. Prior works such as (Das et al., 2024) conduct empirical experiments on LLM alignment tasks.
- The non-linearity of latent reward function is one of the advantage of this paper, but I am not sure how this paper theoretically or empirically support this? E.g. larger depth $D$ leads to better sub-optimality gap?

**Questions:**

Apart from questions in the  weakness section, I have a few more questions:
- Why the penalty $\lambda$ in  (Das et al., 2024) is given explicitly while this paper do not?

---

### Official Review · Reviewer_xgZn · 2025-10-31

**Soundness:** 3
**Presentation:** 2
**Contribution:** 3
**Rating:** 6
**Confidence:** 4

**Summary:**

This paper studies the problem of collecting human preference feedback efficiently in settings where obtaining such feedback is costly. Prior work on active preference-based learning in contextual dueling bandits typically assumes linear reward functions, which limits applicability in real-world systems where reward structures are often non-linear. To address this, the authors propose Neural-ADB, an active learning algorithm built upon the neural contextual dueling bandit framework. Neural-ADB uses a neural network to estimate the latent reward function of context–arm pairs and actively chooses both which contexts to explore and which pairs of arms to query.

The paper’s main contributions are:

Algorithmic: Neural-ADB integrates neural reward modeling with (i) context selection based on maximizing uncertainty and (ii) arm selection strategies based on Upper Confidence Bounds (UCB) and Thompson Sampling (TS).

Theoretical: The authors prove that the worst-case sub-optimality gap of the learned policy decays at rate Ō(√(d̃/T)), improving upon previous methods by a factor of Ō((d̃ log T)^{1/2}). They also provide a new bound on the maximum Mahalanobis norm of vectors with respect to a Gram matrix formed from adaptive samples.

Empirical: Experiments on synthetic (Square and Sine) datasets show improved sample-efficiency and lower suboptimality gap compared to baseline active dueling bandit algorithms, including neuralized variants of prior work.

**Strengths:**

Addresses a practical modeling gap. Moving from linear to non-linear latent reward representations makes the framework more suitable for real-world tasks, including recommendation and RLHF-like settings.

Non-trivial theoretical contribution. The analysis of the Gram matrix built from adaptively chosen samples and the resulting suboptimality guarantees are technically interesting and appear novel.

Clear algorithmic structure. The distinction between context selection and arm selection is well-motivated, and both components are presented in a coherent and implementable way.

Empirical evidence aligns with theory. The synthetic experiments consistently demonstrate lower suboptimality and regret compared to linear and non-active baselines.

**Weaknesses:**

Literature positioning could be stronger. The paper situates itself primarily within recent active dueling bandit approaches, but there is a broader body of work on preference elicitation and query selection strategies—including approaches grounded in optimal experimental design—that appear conceptually related. More explicit contextualization would help clarify where Neural-ADB’s contributions sit in relation to earlier frameworks that also optimize which comparisons to request in order to reduce uncertainty.

Experiments are limited in scope. While synthetic tasks are suitable for controlled evaluation, it remains unclear how the method behaves in practical systems where contexts may be structured, annotators may be noisy or inconsistent, and feature representations may evolve.

Computational overhead is not discussed. The algorithm retrains a neural network repeatedly, which may be costly at scale. Some analysis or reporting of training/runtime characteristics would strengthen the practical relevance.

Hyperparameter and model design choices are not deeply examined. The method’s sensitivity to neural architecture, optimizer settings, or initialization strategy is unclear, and robustness across these choices is not empirically explored.

**Questions:**

None at the moment.

---

### Meta-Review · Area_Chair_J1Uq · 2026-01-01

**Summary:**

This paper proposes a dueling bandit algorithm where the generalization model is a neural network. The algorithm is analyzed, and has a better worst-case suboptimality gap rate than Mehta et al. (2023) and Das et al. (2024). The algorithm is empirically evaluated on synthetic datasets. The problem is well motivated, the paper is well written, the algorithm is clearly presented, and the synthetic experiments are well executed. The main concerns of the reviewers are:

* **Novelty:** The paper builds on many prior techniques and the technical novelty is unclear. The authors need to discuss in-depth similarities and differences with dueling, kernel, and neural bandits. In addition, the authors should discuss recent works based on optimal designs. See [Optimal Design for Human Preference Elicitation](https://proceedings.neurips.cc/paper_files/paper/2024/hash/a40ff56daab9f4808b1e18350c8a11ce-Abstract-Conference.html) and references in this paper.

* **Correctness:** The proof of Theorem 1 is incorrect. In particular, one lower bound is actually an upper bound, and this breaks the proof.

* **Experiments:** All experiments are synthetic. Therefore, the real-world performance of the proposed algorithm is unclear. The computational cost of the algorithm should also be compared to linear approaches. The expected tradeoff is that the linear approaches are more efficient but less accurate.

There was no rebuttal. Since the correctness is questioned and the technical novelty is unclear, this paper requires a major revision and cannot be accepted at this time.

**Reviewer Concerns:**

There was no rebuttal.

**Reviewer Scores:**

There was no rebuttal.

---

### Decision · Program_Chairs · 2026-01-26

Reject